# Acidic pH and divalent cation sensing by PhoQ are dispensable for systemic salmonellae virulence

**Kevin G Hicks[1], Scott P Delbecq[2], Enea Sancho-Vaello[3], Marie-Pierre Blanc[1], Katja K Dove[2], Lynne R Prost[4], Margaret E Daley[5], Kornelius Zeth[6,7], Rachel E Klevit[2], Samuel I Miller[1,8,9]***

[1]Department of Microbiology, University of Washington Medical School, Seattle, United States; [2]Department of Biochemistry, University of Washington Medical School, Seattle, United States; [3]Unidad de Biofisica, Centro Mixto Consejo Superior de Investigaciones Cientificas-Universidad del País Vasco/Euskal Herriko Unibertsi-tatea (CSIC,UPV/EHU), Leioa, Bizkaia, Spain; [4]Department of Biochemistry, University of Wisconsin–Madison, Madison, United States; [5]Department of Chemistry and Biochemistry, University of San Diego, San Diego, United States; [6]Department of Biochemistry and Molecular Biology, University of Basque Country, Leioa, Spain; [7]IKERBASQUE, Basque Research Organisation for Science, Bilbao, Spain; [8]Department of Genome Sciences, University of Washington Medical School, Seattle, United States; [9]Department of Medicine, University of Washington Medical School, Seattle, United States

**Abstract** *Salmonella* PhoQ is a histidine kinase with a periplasmic sensor domain (PD) that promotes virulence by detecting the macrophage phagosome. PhoQ activity is repressed by divalent cations and induced in environments of acidic pH, limited divalent cations, and cationic antimicrobial peptides (CAMP). Previously, it was unclear which signals are sensed by salmonellae to promote PhoQ-mediated virulence. We defined conformational changes produced in the PhoQ PD on exposure to acidic pH that indicate structural flexibility is induced in α-helices 4 and 5, suggesting this region contributes to pH sensing. Therefore, we engineered a disulfide bond between W104C and A128C in the PhoQ PD that restrains conformational flexibility in α-helices 4 and 5. PhoQ$^{W104C-A128C}$ is responsive to CAMP, but is inhibited for activation by acidic pH and divalent cation limitation. *phoQ*$^{W104C-A128C}$ *Salmonella enterica* Typhimurium is virulent in mice, indicating that acidic pH and divalent cation sensing by PhoQ are dispensable for virulence.

*For correspondence: millersi@uw.edu

**Competing interests:** The authors declare that no competing interests exist.

**Reviewing editor**: Feng Shao, National Institute of Biological Sciences, China

## Introduction

Salmonellae are Gram-negative bacterial pathogens that cause severe gastroenteritis and systemic disease in animals and humans. Critical for salmonellae virulence is their ability to survive and replicate within host cells (*Fields et al., 1986*). Following phagocytosis by macrophages, salmonellae are contained within a phagosomal environment containing a diversity of antimicrobial factors including proteases, reactive oxygen and nitrogen species, acidic pH and cationic antimicrobial peptides (CAMP) (*Flannagan et al., 2009*). Salmonellae have multiple mechanisms, including the PhoQ sensor, to sense the phagosomal milieu and respond by increasing their resistance to host antimicrobial factors (*Haraga et al., 2008*; *Chen and Groisman, 2013*; *Dalebroux and Miller, 2014*). PhoQ is the sensor kinase component of the PhoPQ two-component regulatory system that governs the

**eLife digest** *Salmonella* bacteria cause illnesses in humans, such as food poisoning and typhoid fever. In response to a *Salmonella* infection, immune cells known as macrophages detect and engulf the bacteria. The conditions inside the macrophage (which include an acidic pH and high levels of antimicrobial molecules) can destroy some bacteria. However, *Salmonella* bacteria (which are also called salmonellae) can sense and counteract these hostile conditions; this allows them to remodel their surface to survive and reproduce inside macrophages and continue to cause disease.

A protein known as PhoQ, which is found on the surface of *Salmonella* bacteria, is a sensor that detects when the bacterium is inside a macrophage and so needs to boost its defenses. The PhoQ sensor is able to respond to acidity, the absence of divalent cations—such as magnesium and calcium ions—and certain antimicrobial peptide molecules. These conditions and components are used inside macrophages to try and kill the bacteria, but it was not known which of these signals PhoQ actually senses during an infection.

Hicks et al. established how the sensor region of PhoQ changes when it is exposed to acid. This knowledge enabled variants of this protein to be constructed that do not respond when exposed to acidic conditions or low levels of divalent cations. Salmonellae that have these modified PhoQ sensors were still able to infect macrophages and cause disease in mice. These findings suggest that antimicrobial peptide sensing alone is sufficient to trigger the bacteria's defenses inside host organisms.

Understanding how salmonellae detect antimicrobial factors could help with the development of new treatments for the diseases caused by these bacteria. Furthermore, the new tools developed by Hicks et al. could be applied to other systems to characterize how bacteria interact with their host environment during infection.

phosphorylated state of the response regulator PhoP (*Groisman et al., 1989*; *Miller et al., 1989*). PhoQ exists as a dimer within the inner membrane and has a periplasmic sensor domain (PD) that transduces signals across the inner membrane to the cytoplasmic histidine kinase domain. Following activation of PhoQ by the phagosomal environment, PhoP is phosphorylated and transcriptionally controls a large network of genes (>300), many of which are involved in virulence (*Fields et al., 1989*; *Behlau and Miller, 1993*; *Belden and Miller, 1994*; *Gunn and Miller, 1996*; *Guo et al., 1997*; *Bearson et al., 1998*; *Guo et al., 1998*; *Adams et al., 2001*; *Bader et al., 2003*; *Dalebroux et al., 2014*). Precise PhoPQ-mediated gene regulation is essential for salmonellae infection as strains with null or constitutively active mutations in PhoPQ are highly attenuated for virulence in animals and humans (*Fields et al., 1989*; *Galán and Curtiss, 1989*; *Miller et al., 1989*; *Miller and Mekalanos, 1990*).

The PhoQ PD is a member of the PAS-fold and PDC-fold domain families (*Cho et al., 2006*; *Cheung et al., 2008*; *Cheung and Hendrickson, 2010*). Unlike other PDC-sensors, which bind small ligands in a defined binding pocket or PhoQ PD homologs found in environmental bacteria, the PhoQ PD from bacteria that primarily interact with animals has no apparent binding pocket due to an occluding structural element: α-helices 4 and 5 (*Cho et al., 2006*; *Prost et al., 2007*; *Cheung et al., 2008*; *Prost et al., 2008*). Acidic residues on α4 and α5 and β-strands 5 and 6 in the PhoQ PD form a structural scaffold for binding antimicrobial peptides, as well as the divalent cations $Mg^{2+}$, $Mn^{2+}$, and $Ca^{2+}$ (*Waldburger and Sauer, 1996*; *Bader et al., 2005*; *Cho et al., 2006*; *Prost et al., 2008*). PhoQ kinase activity is repressed and phosphatase activity is dominant at millimolar or greater concentrations of divalent cations (*Garcia Vescovi et al., 1996*; *Castelli et al., 2000*; *Montagne et al., 2001*), presumably due to divalent cation salt-bridges formed between the PD acidic patch and inner membrane phospholipids (*Cho et al., 2006*). Additionally, PhoQ activity is repressed by feedback inhibition involving the small inner membrane protein, MgrB (*Lippa and Goulian, 2009*). Conversely, bacterial growth in sub-millimolar divalent cation conditions results in PhoQ activation and increased kinase activity (*Garcia Vescovi et al., 1996*), presumably due to disruption of salt-bridges between the PhoQ PD and inner membrane. However, the macrophage phagosome has a magnesium concentration of approximately one millimolar and a calcium concentration of approximately 500 micromolar, suggesting PhoQ is not activated by divalent cation limitation during

intracellular infection (*Christensen et al., 2002*; *Martin-Orozco et al., 2006*). At one millimolar divalent cation concentration, PhoQ can be activated by exposure to pH 5.5 or sub-inhibitory concentrations of CAMP (*Bader et al., 2005*; *Prost et al., 2007*). These are relevant host signals as the macrophage phagosome acidifies to approximately pH 5.5 and contains CAMP (*Alpuche Aranda et al., 1992*; *Rathman et al., 1996*; *Rosenberger et al., 2004*; *Martin-Orozco et al., 2006*). Furthermore, neutralization of acidified macrophage intracellular compartments with chemical inhibitors results in decreased PhoQ-mediated gene expression during infection (*Alpuche Aranda et al., 1992*; *Martin-Orozco et al., 2006*). Combined, these findings suggested a model in which acidic pH and CAMP activate PhoQ within the macrophage phagosome; however, the individual contribution of these signals to PhoQ-mediated virulence remained unknown.

Acidic pH and CAMP additively activate PhoQ suggesting that the PD has distinct sensing mechanisms for these stimuli (*Prost et al., 2007*). A variety of experimental data indicate that CAMP directly compete with divalent cations for binding sites within the PhoQ PD acidic patch, leading to a model in which CAMP activates PhoQ by disrupting salt-bridges with the inner membrane (*Bader et al., 2005*). The mechanism by which PhoQ is activated by acidic pH appears to be distinct from CAMP and involves perturbations to a network of residues surrounding H157 within the α/β-core of the PD (*Prost et al., 2007*). In this study, we defined conformational changes that occur within the PhoQ PD on exposure to acidic pH. Characterization of the conformational changes induced by acidic pH inspired the construction of PhoQ variants which are impaired for acidic pH and divalent cation sensing, but retain their ability to respond to CAMP. Prior to this study, it was unclear which signals were important for PhoQ-mediated virulence. Utilizing these PhoQ variants, we have now established that acidic pH and divalent cation sensing are dispensable signals for PhoQ-mediated systemic virulence of *Salmonella enterica* Typhimurium, suggesting that CAMP or other host molecules facilitate PhoQ-dependent pathogenesis.

## Results

### Residues in the PhoQ PD that are dynamic during pH-titration localize proximal to the interface between α-helices 4 and 5 and the α/β-core

PhoQ is activated in acidic conditions in vitro and within the acidified environment of the *Salmonella*-containing vacuole (SCV) after phagocytosis (*Alpuche Aranda et al., 1992*; *Martin-Orozco et al., 2006*; *Prost et al., 2007*). However, the mechanism by which the PhoQ PD senses acidic pH is not well characterized. Previously, we reported that the ($^1$H, $^{15}$N)-HSQC-NMR spectrum of the *S. enterica* Typhimurium PhoQ PD is highly sensitive to changes in pH (*Prost et al., 2007*). Therefore, to further understand PhoQ dynamics during activation by acidic pH we collected a series of ($^1$H, $^{15}$N)-HSQC-NMR spectra of the PhoQ PD as a function of pH (*Figure 1A*). To extract residue information from the spectra, resonance assignments were determined for the PhoQ PD at pH 3.5, the condition that yielded the greatest number of observable resonances. Of the 138 residues that can yield HSQC signals, 120 resonances could be assigned in the spectrum at pH 3.5 (*Figure 1B*). The remarkably well dispersed spectrum indicates that the PD remains stably folded, even at pH of 3.5.

In the absence of other pH-dependent processes, resonances that arise from residues that undergo a protonation/deprotonation event will shift in a continuous manner. Such processes will appear in the so-called 'fast-exchange' NMR regime due to the rapid on/off rate of protons. Many resonances in the PD spectra exhibited pH-dependent fast-exchange behavior, consistent with ionization of the many histidine and acidic residues. In addition, some resonances broadened and disappeared from the spectrum as a function of pH. This behavior corresponds to intermediate-to-slow exchange and is indicative of a conformational change or the existence of multiple states that interconvert slowly. Thus, pH-dependent changes in the PhoQ PD HSQC spectra reveal regions of the domain that experience changes in functional group ionization and conformational dynamics.

Spectra, collected at pH 3.5 and pH 6.5, were compared to identify regions in the PhoQ PD that are sensitive to changes in pH (*Figure 2A*). Resonances that experienced significant pH-dependent chemical shift perturbations (CSPs > 0.08 ppm) or broadened beyond detection, localize to regions of the protein that contain ionizable functional groups and/or experience conformational dynamics; thereby defining pH-responsive regions in the domain. Of the 120 assigned residues in the PhoQ PD, resonances from 42 residues were affected by transition from pH 6.5 to 3.5 (*Figure 2B*). Due to resonance overlap and broadening, it is difficult to partition the two spectroscopic effects throughout

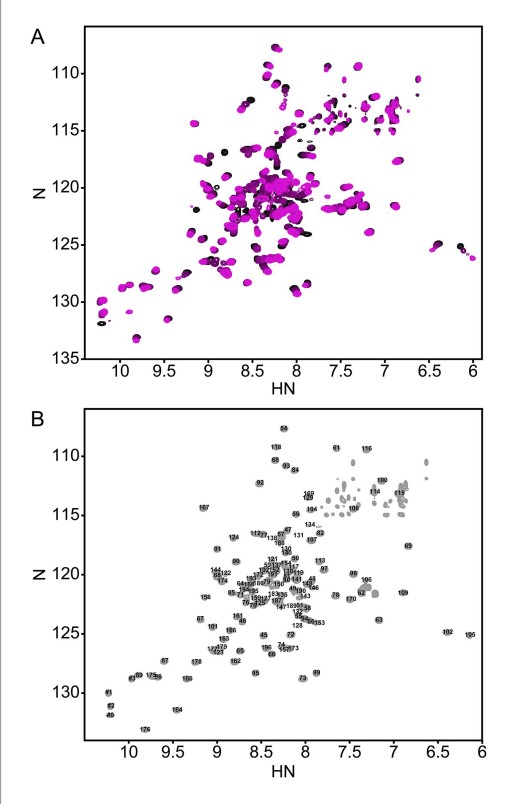

**Figure 1**. The annotated PhoQ PD ($^1$H, $^{15}$N)-HSQC-NMR spectrum reveals significant peak shifting and broadening during pH titration. (**A**) ($^1$H, $^{15}$N)-HSQC-NMR spectra of neutral to acidic pH-titration of the PhoQ PD. The pH-titration is represented as a magenta (pH 6.5) to black (pH 3.5) color gradient. The pH-titration spectra include pH 6.5, 6.0, 5.5, 4.9, 4.1, and 3.5. (**B**) The assigned ($^1$H, $^{15}$N)-HSQC-NMR spectra of the *S. enterica* Typhimurium PhoQ PD at pH 3.5. Residue numbers are labeled proximal to their corresponding peak.

the comparison. Approximately, 20 affected resonances broadened beyond detection at pH 6.5, consistent with pH-dependent conformational dynamics in the PhoQ PD. Furthermore, 66 resonances were relatively unaffected, indicating that the PhoQ PD has pH-insensitive regions.

Assignments for the HSQC spectrum allowed us to identify residues in the PhoQ PD structure that experience pH-dependent changes (*Figure 2C*). A majority of residues affected by pH localize to α1, α2, α4, and α5 and proximal regions, including β5, β6, and β7. As an independent approach, we randomly mutagenized the PD to identify mutations that activate PhoQ in the presence of repressing concentrations of divalent cations (*Figure 2—figure supplement 1*). pH-sensitive regions identified in the NMR experiments overlap or are proximal to many of the mutations identified in our screen for activating mutations in the PhoQ PD (*Figure 2B*, asterisks). Similar to the activating mutations, a majority of the pH-sensitive residues form an interconnected network which spans α4 and α5 and the α/β-core (*Figure 2D*). These data suggested that PhoQ PD residues and structural features important for activation and repression undergo conformational change during pH titration. A majority of the residues that were unaffected by changes in pH mapped distally to α4 and α5, providing support for the hypothesis that the detection and response to pH is contained within localized structural elements of the PD. Altogether, these observations are consistent with a model where fluctuations in pH promote local conformational dynamics between α4 and α5 and the α/β-core as part of the pH sensing mechanism.

## A disulfide bond between α-helices 2 and 4 within the PhoQ PD inhibits activation by acidic pH and divalent cation limitation, but does not restrict activation by CAMP

The NMR and mutagenesis data suggested that α4 and α5 are dynamic and that the relationship between the two helices and the α/β-core of the PD likely plays a role in activation. To test this hypothesis, we engineered a PhoQ PD mutant that contains an internal disulfide bond predicted to restrict conformational dynamics between α4 and α5 and the α/β-core. Cysteine residues were introduced at positions W104 (on α2) and A128 (on α4) based on their side-chain surface exposure, relative geometries, and C$^β$ distance (~6 Å) observed in the *S. enterica* Typhimurium PhoQ PD structure (PDB 1YAX). Non-reducing SDS-PAGE and western blotting of membranes harvested from *phoQ*$^{W104C-A128C}$ *S. enterica* Typhimurium revealed a faster migrating PhoQ species relative to wild type, suggesting W104C and A128C form an intramolecular disulfide bond when expressed in bacteria (*Figure 3—figure supplement 1A*). When treated with sample buffer supplemented with β-mercaptoethanol to reduce the disulfide bond, PhoQ$^{W104C\ A128C}$ migrated similarly to the wild-type protein. Membranes harvested from *phoQ*$^{W104C-A128C}$ *S. enterica* Typhimurium grown in N-minimal media (N-mm) at pH 7.5 or pH 5.5, supplemented with 10 micromolar or 1 millimolar

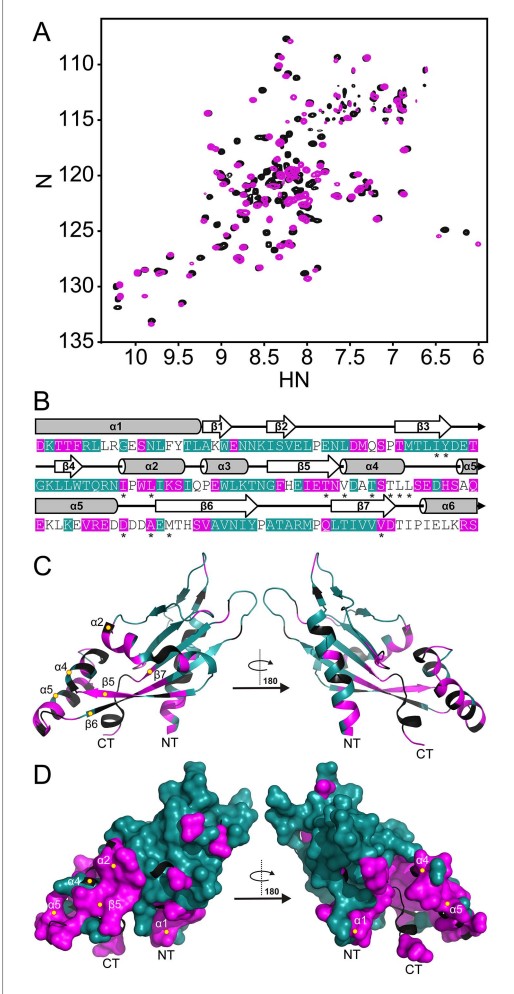

**Figure 2**. The PhoQ PD experiences significant pH-dependent perturbations which map to α4 and α5 and the α/β-core. (**A**) Comparison of ($^1$H, $^{15}$N)-HSQC-NMR spectra of the PhoQ PD at pH 6.5 (magenta) and pH 3.5 (black). (**B**) Residues that experience CSPs >0.08 ppm and/or peak broadening determined from the spectral comparison in panel **A** are mapped onto the *S. enterica* Typhimurium PhoQ PD (residues 45–188) primary and secondary structures (pH-sensitive residues, magenta; pH-insensitive residues, teal; ambiguous or non-assigned residues, no color). The locations of activating mutations from *Figure 2—figure supplement 1* are indicated with asterisks. (**C**) pH-sensitive residues from panel **A** mapped onto the PhoQ PD structure (PDB 1YAX); pH-sensitive residues (magenta), pH-insensitive residues (teal), and ambiguous or non-assigned residues (black). pH-sensitive secondary structural features are labeled with yellow circles (NT, N-termini; CT, C-termini). (**D**) Continuous surface representation (1.4 Å probe) of pH-sensitive (magenta) and pH-insensitive (teal) residues from panel **C** mapped onto the PhoQ PD crystal structure.

The following figure supplement is available for figure 2:

MgCl$_2$, or CAMP showed no observable differences in PhoQ$^{W104C-A128C}$ disulfide bond formation by SDS-PAGE, suggesting formation of the W104C-A128C disulfide bond is not dependent on growth conditions or PhoQ activation state (data not shown). These data indicated that the disulfide bond formed between W104C-A128C is stably maintained within the *S. enterica* Typhimurium periplasm.

The PhoQ$^{W104C-A128C}$ disulfide mutant was designed to inhibit motion between α2 and α4, allowing us to determine whether the dynamics of α4 and α5 play a critical role in activation. When exposed to acidic pH or low divalent cation growth media, activation of the PhoQ-dependent *phoN::TnphoA* reporter in *S. enterica* Typhimurium was significantly reduced in *phoQ$^{W104C-A128C}$* relative to wild type (*Figure 3A*). Additionally, the previously identified T48I activating mutation in the T48 D179 K186 (TDK) network in the PhoQ PD (*Miller and Mekalanos, 1990*; *Garcia Vescovi et al., 1996*; *Sanowar et al., 2003*; *Cho et al., 2006*) was suppressed by the W104C-A128C disulfide bond, supporting the hypothesis that α4 and α5 and the TDK network are an interconnected signaling element (*Figure 3B*). Interestingly, the T48I mutation potentiates CAMP activation in the *phoQ$^{T48I\ W104C-A128C}$* background by an unknown mechanism. Importantly, CAMP still activated the *phoN::TnphoA* reporter in *phoQ$^{W104C-A128C}$* and *phoQ$^{T48I\ W104C-A128C}$* *S. enterica* Typhimurium at or above wild-type levels, indicating that these mutant proteins are functional (*Figure 3A,B*). Chromosomal *phoQ$^{W104C-A128C}$* had a similar phenotype to *phoQ$^{W104C-A128C}$* expressed from the pBAD24 vector, indicating the phenotype is not an artifact of expression *in trans* (*Figure 3—figure supplement 1B*). Furthermore, the *phoQ$^{W104C-A128C}$* phenotype does not appear to be exclusive to *phoN* as other PhoQ-regulated genes—*pagD*, *pagO*, and *phoP*—are significantly reduced for induction by acidic pH and divalent cation limitation, but are induced by exposure to CAMP, similar to wild-type bacteria (*Figure 3—figure supplement 2*). Serine substitutions at W104 and A128 did not recapitulate the phenotype observed for *phoQ$^{W104C-A128C}$*, but rather resulted in increased *phoN::TnphoA* reporter activity relative to wild type (*Figure 3—figure supplement 1C*). Additionally, neither single cysteine nor single serine substitutions at W104 or A128 recapitulated the *phoQ$^{W104C-A128C}$* phenotype (*Figure 3—figure supplement 1D*). These results confirmed that a disulfide bond is required for the *phoQ$^{W104C-A128C}$*

*Figure 2. Continued*

**Figure supplement 1**. Residues involved in PhoQ activation and repression form a buried network connecting α4 and α5 to the α/β-core.

phenotype. Similar to residues identified in our screen for activating mutations, the replacement of partially buried, hydrophobic residues at positions W104 and A128 with smaller, polar side-chains promoted activation. Altogether, these data confirmed that restricting conformational flexibility or movement of α4 and α5 inhibits activation by acidic pH, divalent cation limitation, and activating mutations in the TDK network. These data suggest that CAMP activates PhoQ by a mechanism that is distinct and separable from the mechanism by which acidic pH or divalent cation limitation activate PhoQ.

## The PhoQ^W104C-A128C PD is structurally similar to wild type and has increased stability

A disulfide bond spanning helices α2 and α4 inhibits activation of PhoQ^W104C-A128C by acidic pH and divalent cation limitation. Given the remarkable phenotype of this mutant, we sought to ascertain whether the PhoQ^W104C-A128C PD maintains a similar structure to the wild-type PD. A crystal structure of the *S. enterica* Typhimurium PhoQ^W104C-A128C PD (PDB 4UEY) was solved at 1.9 Å resolution (*Figure 4A* and *Table 1*). As predicted, the PhoQ^W104C-A128C PD formed an intramolecular disulfide bond between W104C and A128C, covalently linking α2 and α4 (*Figure 4A*, inset). The protomers in the PhoQ^W104C-A128C PD structure are highly similar to each other, with an average root mean squared deviation (r.m.s.d.) of 0.3. Furthermore, the disulfide mutant structure is similar to previously solved structures of wild-type *S. enterica* Typhimurium (PDB 1YAX) and *Escherichia coli* (PDB 3BQ8) PhoQ PD, with an average r.m.s.d. value of 1.07 Å (*Figure 4B*). These data further demonstrate that the PhoQ^W104C-A128C PD forms an intramolecular disulfide bond and a structure similar to the wild-type PhoQ PD.

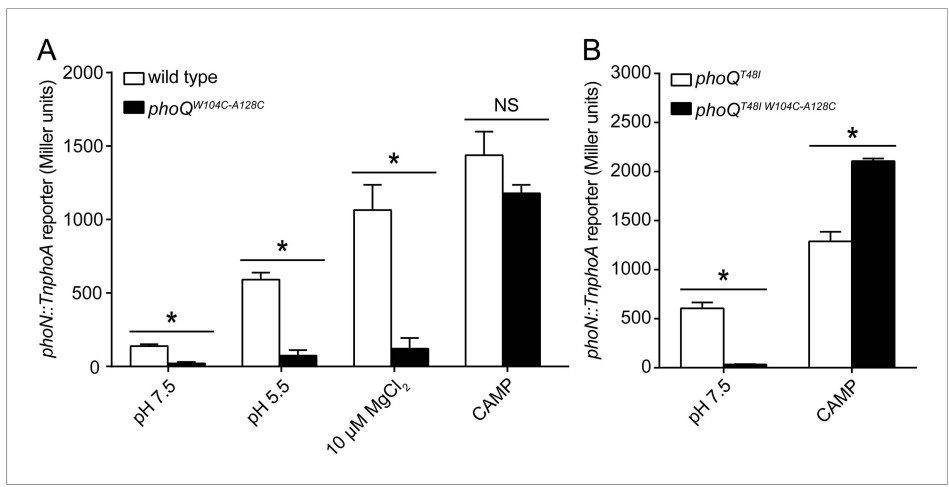

**Figure 3**. A disulfide bond between α-helices 2 and 4 inhibits PhoQ activation by acidic pH and divalent cation limitation, but does not inhibit activation by CAMP. PhoQ-dependent *phoN::TnphoA* alkaline phosphatase activity of (**A**) wild-type and *phoQ*^W104C-A128C or (**B**) *phoQ*^T48I and *phoQ*^T48I W104C-A128C *S. enterica* Typhimurium strains grown in basal (pH 7.5) or activating (pH 5.5, 10 μM MgCl$_2$, or CAMP) N-mm. (**A** and **B**) The data shown are representatives from at least three independent experiments performed in duplicate and presented as the mean ± SD. Unpaired Student's *t*-test was performed between wild type and *phoQ*^W104C-A128C or *phoQ*^T48I and *phoQ*^T48I W104C-A128C for all conditions; (*) $p \leq 0.05$, (NS) not significantly different.

The following figure supplements are available for figure 3:

**Figure supplement 1**. The PhoQ^W104C-A128C disulfide forms in the *Salmonella* periplasm and individual mutations at W104 or A128 do not inhibit activation by acidic pH or divalent cation limitation.

**Figure supplement 2**. Multiple PhoQ-dependent genes in *phoQ*^W104C-A128C *Salmonella* are induced by CAMP, but not by acidic pH or divalent cation limitation.

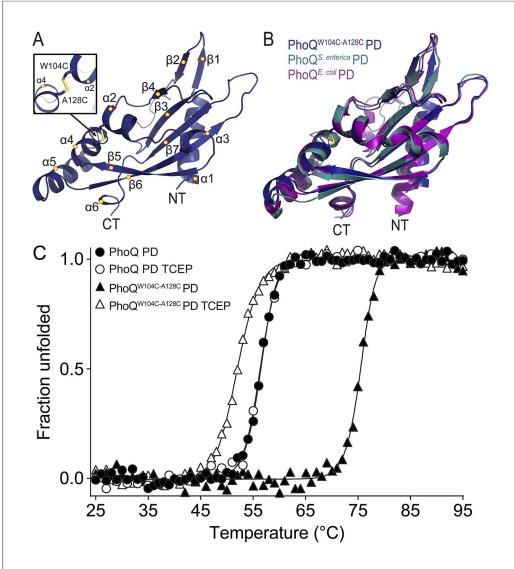

**Figure 4**. The PhoQ$^{W104C\text{-}A128C}$ PD is structurally similar to wild type and has increased stability. (**A**) 1.9 Å crystal structure of the *S. enterica* Typhimurium PhoQ$^{W104C\text{-}A128C}$ PD (PDB 4UEY). The W104C-A128C disulfide bond (inset) is located between α2 and α4. Secondary structural features are annotated with yellow circles (NT, N-termini; CT, C-termini). (**B**) Structural comparison of the PhoQ$^{W104C\text{-}A128C}$ PD (blue), the wild-type *S. enterica* Typhimurium PhoQ PD (PDB 1YAX, teal), and the wild-type *E. coli* PhoQ PD (PDB 3BQ8, purple). (**C**) Thermal denaturation of wild-type *S. enterica* Typhimurium PhoQ PD and PhoQ$^{W104C\text{-}A128C}$ PD treated with or without TCEP reducing agent monitored by CD spectroscopy at 212 nm. Raw data were normalized to give the fraction unfolded protein assuming a two-state denaturation process. A sigmoidal curve was fit to the processed data. The data shown are representatives from three independent experiments.

The following figure supplement is available for figure 4:

**Figure supplement 1**. Wild-type and PhoQ$^{W104C\text{-}A128C}$ PD have similar secondary structure.

We hypothesize that the W104C-A128C disulfide may stabilize conformational dynamics between α4 and α5 and the α/β-core, preventing acidic pH from promoting a flexible, active state. This hypothesis was tested by performing thermal melts on purified PhoQ PD and PhoQ$^{W104C\text{-}A128C}$ PD at pH 5.5 by following the CD signal of each protein as a function of temperature. We first confirmed that purified PhoQ$^{W104C\text{-}A128C}$ PD forms a disulfide as visualized as a shift in SDS-PAGE migration rate relative to PhoQ PD and TCEP-reduced PhoQ$^{W104C\text{-}A128C}$ (*Figure 4—figure supplement 1A*). The CD spectra revealed that the PhoQ PD and PhoQ$^{W104C\text{-}A128C}$ PD with or without TCEP are folded and have relatively similar secondary structure at pH 5.5 and 25°C (*Figure 4—figure supplement 1B*). Thermal denaturation of the PhoQ PD at pH 5.5 proved to be irreversible. Therefore, we reported the apparent transition temperatures ($T_m^{app}$). While the wild-type PhoQ PD unfolded with a $T_m^{app}$ of 56°C in the presence and absence of TCEP (*Figure 4C*), the PhoQ$^{W104C\text{-}A128C}$ PD had a significantly increased $T_m^{app}$ of 75°C. When reduced with TCEP, the PhoQ$^{W104C\text{-}A128C}$ PD was slightly destabilized relative to the wild type, with a $T_m^{app}$ 52°C. Therefore, the W104C-A128C disulfide increased the intrinsic stability of the PD at pH 5.5 relative to wild type. Furthermore, the observations that reduced PhoQ$^{W104C\ A128C}$ PD is less stable than wild type and that *phoQ$^{W104S\ A128S}$* bacteria had increased PhoQ-dependent gene reporter activity relative to wild type (*Figure 3—figure supplement 1C*) suggests that substituting small polar side-chains at these positions in the PD results in decreased stability and increased PhoQ activity. Combined, these results suggest that the mechanism by which the W104C-A128C disulfide bond inhibits PhoQ activation by acidic pH and divalent cation limitation involves a loss of conformational flexibility between α4 and α5 and the α/β-core.

## *Salmonella* strains with the *phoQ$^{W104C\text{-}A128C}$* allele are competent for survival during systemic virulence in mice and within cultured macrophage

Prior to this study, the contribution of specific stimuli to PhoQ-mediated bacterial virulence was difficult to ascertain as mutants that only respond to individual signals were not available. With the construction of *phoQ$^{W104C\text{-}A128C}$* *S. enterica* Typhimurium, the significance of acidic pH and divalent cation sensing by PhoQ to virulence could be directly determined independently of CAMP sensing. Thus, BALB/c mice were infected by the intraperitoneal (IP) route with wild-type, *phoQ$^{W104C\text{-}A128C}$*, or *phoQ* null (*ΔphoQ*) bacteria and splenic bacterial burden was determined at 48- and 96-hpi (*Figure 5A*, solid lines). Similar to infection with wild-type bacteria, mice infected with *phoQ$^{W104C\text{-}A128C}$* bacteria had increased splenic bacterial burden relative to those infected with *ΔphoQ* bacteria. The equivalent experiment was performed in resistant A/J mice to determine if the virulence phenotype observed for mice infected with *phoQ$^{W104C\text{-}A128C}$* bacteria was due to the susceptible BALB/c mouse

**Table 1.** Crystallographic data collection and refinement

| | PhoQ[W104C-A128C] PD |
|---|---|
| Data collection | |
| Space group | C2 |
| Cell dimensions | |
| $a, b, c$ (Å) | 128.04, 45.37, 81.37 |
| $\alpha, \beta, \gamma$ (°) | 90, 102.53, 90 |
| Resolution (Å) | 31.3–1.9 (2.01–1.90)* |
| $R_{sym}$ or $R_{merge}$ | 0.05 (0.51) |
| $I/\sigma I$ | 12.9 (1.6) |
| Completeness (%) | 97.8 (93.8) |
| Redundancy | 3.6 (3.2) |
| Refinement | |
| Resolution (Å) | 31.3–1.90 (1.95–1.90)* |
| No. reflections | 35,633 |
| $R_{work}/R_{free}$ | 0.23/0.26 (0.38/0.44) |
| No. atoms (all) | |
| Protein | 3391 |
| Water | 138 |
| $Ca^{2+}$ | – |
| B-factors | |
| Protein | 44.8 |
| Water | 40.6 |
| R.m.s. deviations | |
| Bond lengths (Å) | 0.007 |
| Bond angles (°) | 1.2 |
| Ramachandran statistics | |
| Residues in favored region no (%) | 409 (98.3) |
| Residues in allowed region no (%) | 7 (1.7) |
| Residues in outlier region no (%) | 0 (0) |
| PDB-entry | 4UEY |
| Crystallization conditions | 0.1 M Bis-Tris pH 6.5, 200 mM $MgCl_2$, 25% Peg3350 |

*Values in parentheses are for highest-resolution shell.

genetic background and to determine whether a subtle fitness defect would be exposed on infection of a relatively resistant inbred mice. Infecting A/J mice revealed the same relative phenotypes for wild-type, phoQ[W104C-A128C], and ΔphoQ bacteria, although, as expected, bacterial burden was lower compared to infected BALB/c mice (**Figure 5A**, dotted lines). These results indicate that PhoQ sensing of acidic pH and divalent cation limitation are dispensable for systemic virulence of *S. enterica* Typhimurium in susceptible and relatively resistant inbred mice.

The importance of PhoQ activation by acidic pH and divalent cation limitation for systemic infection was also assessed by competing phoQ[W104C-A128C] bacteria with wild-type *S. enterica* Typhimurium in IP or peroral (PO) infections of BALB/c mice. The splenic bacterial competitive index (CI) for wild-type, phoQ[W104C-A128C], and ΔphoQ bacteria was determined for both IP and PO infections at 48-hpi or 96-hpi, respectively (**Figure 5B** and **Figure 5—figure supplement 1**). Consistent with the single strain infections, phoQ[W104C-A128C] demonstrated no reduction in CI and, in contrast, was more competitive than wild type. ΔphoQ showed the expected reduction in CI. Altogether, these data indicate that

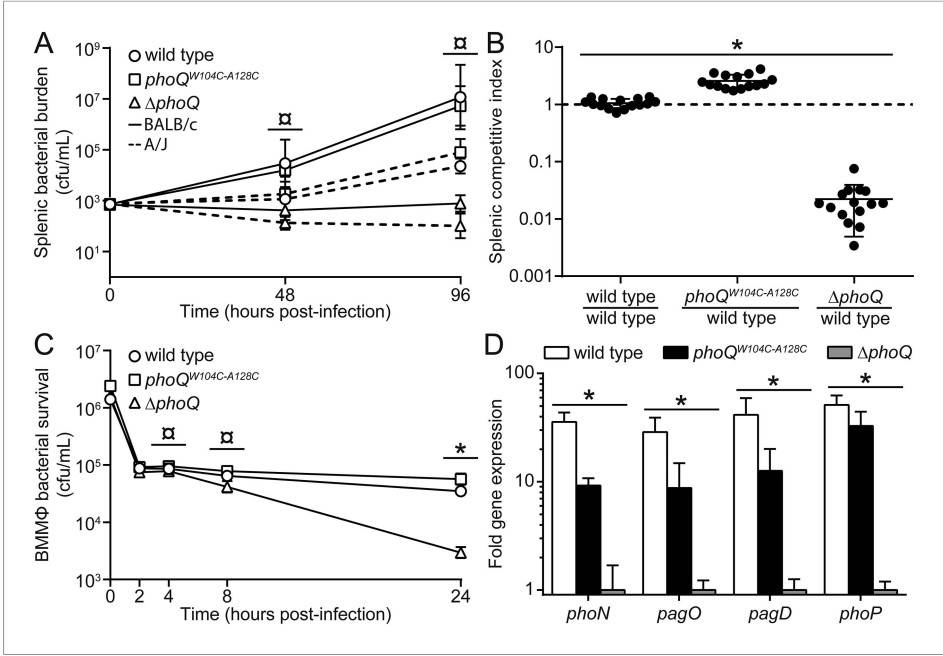

**Figure 5**. *phoQ^{W104C-A128C}* Salmonella survive within host organisms and exhibits PhoQ-dependent gene expression within macrophage. (**A**) Individual *S. enterica* Typhimurium strains administered IP to BALB/c (solid lines) or A/J (dotted lines) mice. The inoculum is shown at T = 0 hpi. Spleens were harvested and bacterial burden quantified. (**B**) Competition between *S. enterica* Typhimurium strains administered IP to BALB/c mice. Spleens were harvested, bacteria quantified 48-hpi and CI determined. (**A** and **B**) The data shown are representatives from at least three independent experiments performed in quintuplet and presented as the mean ± SD. (**C**) BALB/c BMMΦ infected with strains of *S. enterica* Typhimurium. Bacteria were harvested and quantified at the indicated time-points. The inoculum is shown at T = 0 hpi. The data shown are representatives from at least three independent experiments performed in triplicate and presented as the mean ± SD. (**D**) PhoQ-dependent gene expression from *S. enterica* Typhimurium strains within BALB/c BMMΦ 4-hpi. Gene expression was normalized to *rpoD* and presented as fold-induction relative to *ΔphoQ*. The data shown are representatives from at least three independent experiments and presented as the mean ± SD. (**A**, **B**, **C**, and **D**) Unpaired Student's *t*-test was performed between all strains (bar) for each time-point or gene. Symbols for significant difference; (¤) wild type and *phoQ^{W104C-A128C}* are not significantly different from each other ($p \geq 0.05$), but are significantly different from *ΔphoQ* ($p \leq 0.05$), (*) all strains are significantly different from each other ($p \leq 0.05$).

The following figure supplements are available for figure 5:

**Figure supplement 1**. Acidic pH and divalent cation sensing by PhoQ are dispensable for PO systemic competition of *S. enterica* Typhimurium.

**Figure supplement 2**. The in vitro growth rate of wild-type *Salmonella* is decreased relative to *phoQ^{W104C-A128C}* and *ΔphoQ* when grown at pH 5.5.

PhoQ activation by acidic pH and divalent cation limitation are dispensable for *S. enterica* Typhimurium to out compete strains with these capabilities during systemic infection of susceptible mice. Furthermore, the observation that IP and PO administered *phoQ^{W104C-A128C}* *S. enterica* Typhimurium have similar competitive indices suggests that acidic pH and divalent cation sensing by PhoQ are not required for survival or spread from the gastrointestinal tract to deep tissue sites.

*S. enterica* Typhimurium is growth restricted in cultured fibroblasts and nonphagocytic stromal cells in the murine lamina propria via PhoPQ-dependent processes (*Cano et al., 2001*; Nunez-Hernandez et al., 2013). Thus, it was plausible that the competitive advantage observed for *phoQ^{W104C-A128C}* relative to wild-type bacteria was due to an increased growth rate resulting from the loss of acidic pH sensing by PhoQ. When grown in N-mm pH 5.5, wild-type bacterial growth rate was decreased relative to *phoQ^{W104C-A128C}* and *ΔphoQ* (*Figure 5—figure supplement 2*). Conversely, wild

type, $phoQ^{W104C-A128C}$, and $\Delta phoQ$ grown in N-mm pH 7.5 had similar growth kinetics. These data provide evidence that acidic pH sensing by PhoQ reduces *S. enterica* Typhimurium growth rate in vitro and correlates with the in vivo competitive advantage that was observed for $phoQ^{W104C-A128C}$ bacteria within mice spleens.

The contribution of acidic pH and divalent cation limitation as signals for PhoQ-mediated bacterial intracellular survival within macrophages was evaluated by measuring *S. enterica* Typhimurium survival after infection of bone-marrow derived macrophages (BMMΦ) from BALB/c mice. BMMΦ were infected with-wild type, $phoQ^{W104C-A128C}$, or $\Delta phoQ$ *S. enterica* Typhimurium strains and bacterial burden was determined at 2-, 4-, 8-, and 24-hpi (*Figure 5C*). No difference in bacterial burden was observed between wild type, $phoQ^{W104C-A128C}$, and $\Delta phoQ$ at 2- or 4-hpi. At 8- and 24-hpi, bacterial burden for $\Delta phoQ$ was decreased relative to wild type and $phoQ^{W104C-A128C}$. Importantly, bacteria with the $phoQ^{W104C-A128C}$ allele maintained at or above wild-type bacterial levels throughout infection. These data indicate that activation of PhoQ by acidic pH and divalent cation limitation are dispensable for *S. enterica* Typhimurium survival within BMMΦ from inbred mice.

## PhoQ-dependent gene expression is induced in $phoQ^{W104C-A128C}$ *Salmonella* within cultured macrophages

The discovery of the $phoQ^{W104C-A128C}$ phenotype allowed for the unique opportunity to examine the contribution of acidic pH and divalent cation sensing to PhoQ-dependent gene expression during infection of macrophages. Therefore, BMMΦ from BALB/c mice were infected with wild-type, $phoQ^{W104C-A128C}$, or $\Delta phoQ$ *S. enterica* Typhimurium. Following incubation, PhoQ-dependent gene expression was determined for intracellular *S. enterica* Typhimurium (*Figure 5D*). Wild-type bacteria experienced 41-, 29-, 36-, and 51-fold increases in gene expression for *pagD*, *pagO*, *phoN*, and *phoP*, whereas $phoQ^{W104C-A128C}$ experienced increases of 12-, 9-, 9-, and 33-fold, respectively, relative to $\Delta phoQ$ bacteria. These data may indicate that wild-type acidic pH or divalent cation sensing contribute an approximate threefold to fourfold increase in PhoQ-dependent gene expression relative to $phoQ^{W104C-A128C}$; however, a significant amount of gene expression ($\geq$ninefold) appeared to be independent of acidic pH or divalent cation sensing. These findings are consistent with in vitro results which show that acidic pH and CAMP are additive signals for PhoQ (*Prost et al., 2007*). These data indicate that maximal PhoQ-dependent gene expression in macrophages requires acidic pH or divalent cation sensing. Furthermore, these findings reveal that the $phoQ^{W104C-A128C}$ allele promotes significant induction of PhoQ-dependent gene expression, suggesting CAMP or alternative host factors, other than acidic pH and divalent cation limitation, are a major signal for *S. enterica* Typhimurium within BMMΦ vacuoles.

## Discussion

Salmonellae encounter changing environments within the macrophage phagosome and other mammalian host sites during infection. These environments include a variety of antimicrobial factors for which the bacteria must regulate inducible resistance mechanisms in order to survive. These bacterial resistance mechanisms are essential for successful infection, necessitating tight regulation by sensors such as PhoQ. Our study defines α4 and α5 in the PhoQ PD as a pH-responsive structural element that experiences a change in its dynamic behavior upon transition to acidic pH. Furthermore, mutations within the PhoQ PD, predicted to destabilize hydrophobic packing and hydrogen bonding between the α/β-core and α4 and α5, resulted in loss of PhoQ repression. Limiting flexibility between these structural elements by introduction of a disulfide bond inhibited PhoQ activation by acidic pH and divalent cation limitation. We suggest that PhoQ has evolved α4 and α5 as a unique pH-responsive structural element within the PD, effectively replacing the ligand-binding site that is often found in a similar location in other structurally related PDC sensor domains (*Cho et al., 2006*; *Cheung and Hendrickson, 2008*).

This study provides insights for a refined model of PhoQ activation (*Figure 6*). At neutral pH and millimolar divalent cation concentration (*Figure 6*, left), the PhoQ PD is anchored to the inner membrane in a repressed state via cation-bridges, a rigid α/β-core and TDK network, and quiescent α4 and α5. Acidic pH or divalent cation limitation promotes a change in α4 and α5 from a stable to a dynamic state (*Figure 6*, middle). Acidic pH-induced flexibility in α4 and α5 may destabilize divalent cation salt-bridges between the inner membrane and acidic patch, thereby promoting a loss of

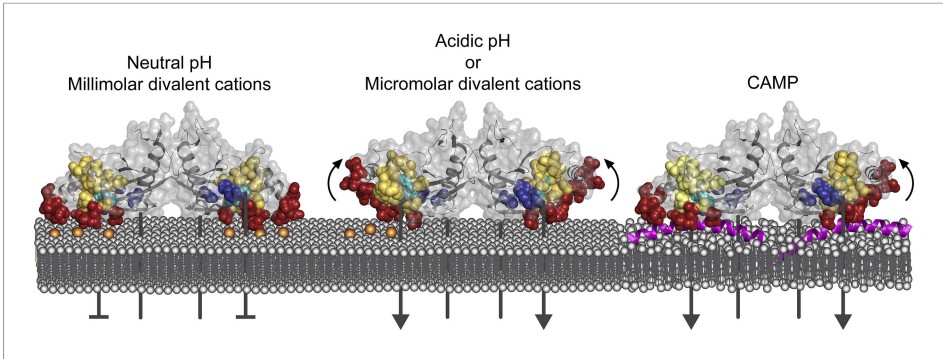

**Figure 6**. Model of PhoQ activation and repression. (Left) At neutral pH and millimolar divalent cation concentration, the PhoQ PD is maintained in a repressed conformation due to rigidified interactions between the α/β-core (yellow spheres), α4 and α5, and salt-bridges (bronze spheres) formed between the acidic patch (red spheres) and inner membrane. (Middle) Transition to a mildly acidic (left protomer) or divalent cation limited (right protomer) environment promotes flexibility in α4 and α5 (bent arrows) and conformational dynamics in the α/β-core surrounding H157 (teal spheres). Movement in α4 and α5 due to acidic pH or divalent cation limitation destabilizes salt-bridges between the acidic patch and inner membrane perturbing the TDK network (blue spheres) resulting in activation. (Right) CAMP (magenta helices) intercalates into the inner membrane and promotes PhoQ activation by directly interacting with the PhoQ transmembrane domains and/or by disrupting local phospholipid packing (left protomer) and/or by overcoming constraints in α4 and α5 (right monomer, bent arrow).

divalent cation-mediated repression. Lack of divalent cation salt-bridges between the PhoQ PD acidic patch and inner membrane due to divalent cation limitation may result in electrostatic repulsion between the acidic patch and inner membrane, releasing α4 and α5, and favoring a more flexible state in this structural element. Changes in the relationship between α4 and α5 and the α/β-core surrounding H157 are transmitted to the dimerization interface and TDK network proximal to the membrane resulting in alterations of the transmembrane domain, cytoplasmic HAMP domain, and ultimately resulting in increased PhoQ kinase activity.

Our model of PhoQ activation and repression has similarities to a recently proposed two-state computational model in which the PD is predicted to experience broad conformational changes within the periplasmic dimerization interface and acidic patch (*Molnar et al., 2014*). This model is consistent with predictions previously made in relation to the discovery of the divalent cation bridges between the PhoQ acidic patch and negatively charged membrane phospholipids (*Bader et al., 2005*; *Cho et al., 2006*). Molnar et al suggest that the PhoQ PD assumes alternative conformations as the acidic patch moves away from the membrane in the absence of divalent cation. Our findings that restricting movement in α4 and α5 inhibits PhoQ activation by acidic pH and divalent cation limitation supports the model that the acidic patch and α4 and α5 must remain dynamic for proper signaling. Furthermore, our observations that PhoQ activation by acidic pH and divalent cation limitation are separable from CAMP-mediated activation may indicate that distinct conformational states exist for each of the unique PhoQ-activating and -repressing stimuli.

Results presented here indicate that α4 and α5 within the PhoQ PD do not adopt a distinct conformation under activating pH conditions, but rather exchange between a rigid, repressed state and an ensemble of conformations that together constitute the acidic pH-activated state. Unlike other pH-sensors that utilize discrete histidine protonation as a mechanism of activation (*Perier et al., 2007*; *Dawson et al., 2009*; *Müller et al., 2009*; *Choi et al., 2013*; *Williamson et al., 2013*), PhoQ activation by acidic pH does not appear to rely strictly on protonation of histidines. Mutagenesis of H157, which is located in the PhoQ PD α/β-core and is observed to form hydrogen bonds with T129 on α4 and the backbone hydroxyl of T180 on the β7-α6 loop, results in a modest increase in activity; however, it does not account for the entire pH-mediated activation state (*Prost et al., 2007*). Mutation of the other two histidines in the PD (H137 and H120) did not significantly affect activation or repression of PhoQ (data not shown). An alternate mechanism of PhoQ activation by acidic pH may involve pH-induced conformational changes within the periplasmic dimerization interface. Notably, an intermolecular disulfide at position T61C in the PhoQ PD dimerization interface results in increased PhoQ-dependent

reporter activity at pH 5.5, suggesting that conformational changes in the dimerization interface can directly affect signal transduction (data not shown). Furthermore, analytical ultracentrifugation analysis revealed that the PhoQ PD dimer dissociates as the pH decreases (data not shown). Additionally, work by Molnar et al support the notion that conformational changes with the dimerization interface are concomitant with activation and repression. Recently, it was shown that the *Helicobacter pylori* chemotaxis receptor, TlpB, utilizes a unique mechanism to sense acidic pH (*Goers Sweeney et al., 2012*). Interestingly, the TlpB PD may sense pH by adopting a 'relaxed' conformation at low pH due to decreased hydrogen bonding to a coordinated urea molecule. It is plausible that similar relaxation may occur in the PhoQ PD between the α/β-core and helices α4 and α5 upon exposure to acidic pH. Therefore, pH-induced conformational changes resulting in structural relaxation or flexibility may be an important mechanism by which pH can be sensed.

Our previous work suggests that PhoQ activation by acidic pH and CAMP proceed via different mechanisms (*Bader et al., 2005*; *Prost et al., 2007*). In this study, we have shown that activation by acidic pH and divalent cation limitation are separable from CAMP-mediated activation by rigidifying the interaction between α4 and α5 and the α/β-core. Perhaps, CAMP circumvents α4 and α5 and activates PhoQ via direct interactions within the transmembrane regions adjacent to the acidic patch or disrupts local phospholipid packing promoting conformational changes in the periplasmic and transmembrane domains (*Figure 6*, right). Alternatively, it is plausible that CAMP functions as a large steric 'wedge'. In this scenario, CAMP is recruited to the acidic patch of PhoQ resulting in conformational changes in the PD overcoming any repressive structural constraints between α4 and α5 and the α/β-core.

Determining the host signals which activate PhoQ has been difficult; past investigations have relied on alterations to host processes via chemical inhibitors to neutralize acidic compartments or targeted mutagenesis to remove known PhoQ-activating antimicrobial peptides from host animals (*Alpuche Aranda et al., 1992*; *Martin-Orozco et al., 2006*; *Richards et al., 2012*). Although informative, these studies do not account for unintended host-cell changes due to chemical neutralization of acidic vesicles and organelles or uncharacterized host peptides or molecules which may activate the system. For example, distinct pH-gradients are established and required in eukaryotic vesicular trafficking pathways and chemical neutralization of acidic compartments within these pathways results in a variety of cellular disturbances including inhibition of acidic hydrolases and proteases, perturbation of molecular sorting and recycling, endocytosis and exocytosis dysregulation, and disruption of vesicular fusion events (*Dean et al., 1984*; *Mellman et al., 1986*; *Casey et al., 2010*). Furthermore, chemical neutralization of the SCV is detrimental to intracellular *S. enterica* Typhimurium as it results in decreased bacterial survival (*Rathman et al., 1996*), presumably due to lack of virulence factor expression. Therefore, it is plausible that the use of chemical inhibitors to block or neutralize acidic host-compartments inhibits processes involved in CAMP maturation or trafficking, thereby preventing activation of PhoQ.

It is plausible that undefined host molecules or conditions, that require defined pH-gradients, activate PhoQ in vivo. Our observations that activation by acidic pH and divalent cation limitation are not required for significant increases in PhoQ-dependent gene expression in BALB/c macrophage and that PhoQ-dependent gene expression is induced in CRAMP-deficient macrophage (*Richards et al., 2012*) indicate that uncharacterized host factors, which are likely to be a variety of different cationic antimicrobial molecules, activate PhoQ. Furthermore, multiple host peptides may activate PhoQ in vivo as various host and synthetically derived cationic peptides can activate the system; this is consistent with the large PhoQ PD acidic patch which likely evolved to bind diverse CAMP (*Bader et al., 2003*, *2005*; *Shprung et al., 2012*). From these observations, it is reasonable to speculate that the acidic patch, located on the α4/α5 structural unit within the PhoQ PD of bacteria that interact with animals, evolved to sense a variety of cationic peptides, as the PhoQ PD from environmental bacteria such as *Pseudomonas aeruginosa* lack these important structural features (*Prost et al., 2008*).

Though it is tempting to speculate that CAMP may be the dominant PhoQ-stimulant during systemic infection, it is important to remember that sensing may be redundant or host-compartment specific. Further experiments will need to be performed to examine the contribution of acidic pH and divalent cation sensing to PhoQ-mediated bacterial survival during transition from the intestinal tract to systemic environments and determine if 'bacterial innate immunity' or the recognition of multiple mammalian signals is redundant. Additionally, perhaps acidic pH and divalent cation sensing by PhoQ are functions required for survival in *ex vivo* environments, beyond animal hosts.

The work in this study led to the construction of a specific *S. enterica* Typhimurium PhoQ mutant that is inhibited for activation by acidic pH or divalent cation limitation. This mutant has wild-type virulence in susceptible and resistant mouse models of systemic infection suggesting that, at least in these models, acidic pH and divalent cation sensing are dispensable for virulence. Although we have shown that the PhoQ[W104C-A128C] disulfide forms in purified proteins and in bacteria grown in culture, we did not measure disulfide formation for W104C-A128C within host tissues. Host compartments replete with strong oxidizing agents, such as the macrophage phagosome, could potentially disrupt disulfide bond formation. However, multiple *Salmonella* virulence factors and homeostatic processes that occur in the macrophage phagosome require disulfide bond formation, indicating that *S. enterica* has robust mechanisms to regulate redox potential and resist hyperoxidation of thiols in the periplasm (*Ellermeier and Slauch, 2004*; *Miki et al., 2004*; *Lippa and Goulian, 2012*). Additionally, we have observed altered PhoQ-dependent gene expression from *phoQ*[W104C-A128C] *S. enterica* Typhimurium within macrophage phagosomes similar to in vitro grown bacteria. Therefore, it is highly likely that efficient formation of the W104C-A128C disulfide bond occurs inside host compartments.

In summary, we provide novel detail to the mechanism by which *S. enterica* Typhimurium PhoQ is activated by acidic pH. We have identified residues and secondary structural elements within the PD which contribute to acidic pH sensing and are important for PhoQ signal transduction. Furthermore, structural studies have led to the engineered bifurcation of PhoQ signaling capabilities; separating acidic pH and divalent cation sensing from CAMP signaling. This discovery has allowed us to determine the contribution of acidic pH and divalent cation sensing to *S. enterica* Typhimurium virulence and will provide valuable insights to the spatial-temporal regulation of PhoQ during pathogenesis.

## Materials and methods

### Bacterial strains and growth conditions

Bacterial strains, plasmids, and primers used in this study can be found in *Tables 2*, *3*. *S. enterica* Typhimurium strain 14028s was the wild-type strain used in this study and all subsequent strains and mutants were derived from this strain. Unless otherwise stated, all alkaline phosphatase activity assays were performed in the CS1081 background with CS1084 as the wild-type control and various alleles of *phoQ* basally expressed from pBAD24. Alkaline phosphatase activity assays were also performed on wild type (KH127) and *phoQ*[W104C-A128C] (KH130) recombined on to the chromosome of CS1081. Bacterial strains were grown in either LB broth or modified N-mm as indicated. Activation of the *phoN::TnphoA* reporter was utilized as previously described (*Bader et al., 2005*; *Prost et al., 2007*). Briefly, bacterial strains were grown overnight in modified N-minimal media pH 7.5 containing 1 mM MgCl$_2$. In the morning, cultures were washed once in the appropriate media and diluted 1:100 in to fresh modified N-minimal media containing either 10 μM, 1 mM, or 10 mM MgCl$_2$ and buffered with either 0.1 M Tris or 0.1 M MES to pH 7.5 or pH 5.5, respectively. Unless stated otherwise, the base growth media is N-minimal media pH 7.5 supplemented with 1 mM MgCl$_2$ and 100 μg•ml$^{-1}$ ampicillin. Following dilution into fresh media, cultures were grown for 5 hr shaking at 37°C. To study *phoN::TnphoA* reporter activation in the presence of CAMP, overnight cultures were washed once in N-minimal media pH 7.5 containing 1 mM MgCl$_2$ and diluted 1:100 into the same growth media. Cultures were then grown to OD$_{600}$ 0.2, treated with 5 μg•ml$^{-1}$ of C18G peptide (Anaspec, Fremont, CA), and grown shaking at 37°C for 90 min. Following incubation, alkaline phosphatase activity was measured. Alkaline phosphatase activity assays were performed according to standard protocol on cultures grown in duplicate and repeated on at least three independent occasions.

### Genetic techniques

All PhoQ alleles with point-mutations were generated on pBAD24-*phoQ* or pET11a-*phoQ* using the appropriate primers pairs (*Table 3*) and a standard site-directed mutagenesis protocol or Gibson assembly (*Gibson et al., 2009*). To generate *phoQ*[W104C-A128C] on the *S. enterica* Typhimurium 14028s chromosome, lambda red allelic exchange methods were utilized (*Gerlach et al., 2007*). Briefly, to engineer *phoQ*[W104C-A128C] on the *S. enterica* Typhimurium chromosome, a tetracycline resistant cassette (*tetRA*) was amplified using primers KH45 and KH46. The resulting *phoQ::tetRA* amplicon was recombined into CS093 generating *phoQ::tetRA* (KH23). Primers KH93 and KH94 were used to amplify *phoQ*[W104C-A128C] from pBAD24-*phoQ*[W104C-A128C] (CS1382). The *phoQ*[W104C-A128C] amplicon was

**Table 2**. Strains and plasmids used in this study

| Strain | Description | Source |
|---|---|---|
| CS093 | 14028s wild type *S. enterica* Typhimurium | ATCC |
| CS1081 | CS093 *phoQ::TPOP phoN::TnphoA* | *Bader et al., 2005* |
| CS1083 | CS1081 pBAD24 | *Bader et al., 2005* |
| CS1084 | CS1081 pBAD24-*phoQ* | *Bader et al., 2005* |
| CS1399 | CS1081 pBAD24-*phoQ*$^{I88N}$ | This work |
| CS1400 | CS1081 pBAD24-*phoQ*$^{Y89N}$ | This work |
| KH45 | CS1081 pBAD24-*phoQ*$^{I102C}$ | This work |
| KH140 | CS1081 pBAD24-*phoQ*$^{L105D}$ | This work |
| CS1402 | CS1081 pBAD24-*phoQ*$^{T124N}$ | This work |
| CS1403 | CS1081 pBAD24-*phoQ*$^{V126E}$ | This work |
| CS1404 | CS1081 pBAD24-*phoQ*$^{T129I}$ | This work |
| CS1405 | CS1081 pBAD24-*phoQ*$^{T131P}$ | This work |
| CS1406 | CS1081 pBAD24-*phoQ*$^{L132P}$ | This work |
| KH28 | CS1081 pBAD24-*phoQ*$^{L133C}$ | This work |
| CS1407 | CS1081 pBAD24-*phoQ*$^{D150G}$ | This work |
| CS1408 | CS1081 pBAD24-*phoQ*$^{A153P}$ | This work |
| CS1409 | CS1081 pBAD24-*phoQ*$^{M155V}$ | This work |
| CS1410 | CS1081 pBAD24-*phoQ*$^{V178D}$ | This work |
| CS1374 | CS1081 pBAD24-*phoQ*$^{W104C}$ | This work |
| CS1386 | CS1081 pBAD24-*phoQ*$^{A128C}$ | This work |
| CS1382 | CS1081 pBAD24-*phoQ*$^{W104C-A128C}$ | This work |
| KH48 | CS1081 pBAD24-*phoQ*$^{W104S}$ | This work |
| KH49 | CS1081 pBAD24-*phoQ*$^{A128S}$ | This work |
| KH50 | CS1081 pBAD24-*phoQ*$^{W104S\ A128S}$ | This work |
| CS1101 | BL21 pET11a-*phoQ* 45-190-(His)$_6$ | *Bader et al., 2005* |
| KH85 | NEB SHuffle T7 express pET11a-*phoQ*$^{W104C-A128C}$ 45-190-(His)$_6$ | This work |
| KH23 | *phoQ::tetRA* | This work |
| KH163 | *phoQ*$^{W104C-A128C}$ | This work |
| CS1350 | *ΔphoQ* | *Prost et al., 2008* |
| KH127 | *phoQ phoN105::TnphoA* | This work |
| KH130 | *phoQ*$^{W104C-A128C}$ *phoN105::TnphoA* | This work |
| KH111 | CS093 pWSK129$^{Kan}$ | This work |
| KH112 | CS093 pWSK29$^{Amp}$ | This work |
| KH113 | *phoQ*$^{W104C-A129C}$ pWSK29$^{Amp}$ | This work |
| KH114 | *ΔphoQ* pWSK29$^{Amp}$ | This work |

recombined into *phoQ::tetra* (KH23). Positive clones for *phoQ*$^{W104C-A128C}$ recombination were identified via Bochner selection (*Bochner et al., 1980*). Chromosomal *phoQ*$^{W104C-A128C}$ was then transduced into a clean *S. enterica* Typhimurium 14028s background via P22 phage transduction. *phoQ*$^{W104C-A128C}$ positive clones were confirmed via DNA sequencing.

To identify residues in the PhoQ PD that when mutated result in increased *phoN::TnphoA* activity, we performed a random mutagenesis screen as previously described (*Cho et al., 2006*). Briefly, pBAD24-*phoQ* was randomly mutagenized using primers LP135 and LP136 to introduce one mutation per 500 bp in the *phoQ PD* using the GeneMorph II EZClone Domain Mutagenesis Kit

**Table 3.** Primer sequences used in this study

| Primer # (name) | Sequence (5′–3′) |
| --- | --- |
| LP135 (RM_Fwd) | CTGGTCGGCTATAGCGTAAGTTTTG |
| LP136 (RM_Rev) | CACGTATACGAACCAGCTCCACAC |
| LP178 (I88N_Fwd) | CGACCATGACGCTGAATTACGATGAAACGG |
| LP179 (I88N_Rev) | CCGTTTCATCGTAATTCAGCGTCATGGTCG |
| LP180 (Y89N_Fwd) | CCATGACGCTGATTAACGATGAAACGGGC |
| LP181 (Y89N_Rev) | GCCCGTTTCATCGTTAATCAGCGTCATGG |
| KH81 (I102C_Fwd) | GACGCAGCGCAACTGTCCCTGGCTGATTAAAAG |
| KH82 (I102C_Rev) | CTTTTAATCAGCCAGGGACAGTTGCGCTGCGTC |
| LP184 (T124N_Fwd) | CTTCCATGAAATTGAAAACAACGTAGACGCCACC |
| LP185 (T124N_Rev) | GGTGGCGTCTACGTTGTTTTCAATTTCATGGAAG |
| LP186 (V126E_Fwd) | GAAATTGAAACCAACGAAGACGCCACCAGCAC |
| LP187 (V126E_Rev) | GTGCTGGTGGCGTCTTCGTTGGTTTCAATTTC |
| LP188 (T129I_Fwd) | CAACGTAGACGCCATCAGCACGCTGTTG |
| LP189 (T129I_Rev) | CAACAGCGTGCTGATGGCGTCTACGTTG |
| KH192 (L105D_Fwd) | GCGCAACATTCCCTGGGATATTAAAAGCATTCAAC |
| KH193 (L105D_Rev) | GTTGAATGCTTTTAATATCCCAGGGAATGTTGCGC |
| LP190 (L131P_Fwd) | CAACGTAGACGCCACCAGCCCACTGTTGAGCGAAGACCATTC |
| LP191 (L131P_Rev) | GAATGGTCTTCGCTCAACAGTGGGCTGGTGGCGTCTACGTTG |
| LP192 (L132P_Fwd) | GACGCCACCAGCACGCCATTGAGCGAAGACCATTC |
| LP193 (L132P_Rev) | GAATGGTCTTCGCTCAATGGCGTGCTGGTGGCGTC |
| KH85 (L133C_Fwd) | CACCAGCACGCTGTGTAGCGAAGACCATTC |
| KH86 (L133C_Rev) | GAATGGTCTTCGCTACACAGCGTGCTGGTG |
| LP194 (D150G_Fwd) | GTACGTGAAGATGGCGATGATGCCGAG |
| LP195 (D150G_Rev) | CTCGGCATCATCGCCATCTTCACGTAC |
| LP196 (A153P_Fwd) | GAAGATGACGATGATCCCGAGATGACCCAC |
| LP197 (A153_Rev) | GTGGGTCATCTCGGGATCATCGTCATCTTC |
| LP198 (M155V_Fwd) | GACGATGATGCCGAGGTAACCCACTCGGTAGC |
| LP199 (M155V_Rev) | GCTACCGAGTGGGTTACCTCGGCATCATCGTC |
| LP200 (V178D_Fwd) | CCATCGTGGTGGACGATACCATTCCG |
| LP201 (V178D_Rev) | CGGAATGGTATCGTCCACCACGATGG |
| LP141 (W104C_Fwd) | GCGCAACATTCCCTGCCTGATTAAAAGCATTC |
| LP142 (W104C_Rev) | GAATGCTTTTAATCAGGCAGGGAATGTTGCGC |
| LP145 (A128C_Fwd) | GAAACCAACGTAGACTGCACCAGCACGCTGTTG |
| LP146 (A128C_Rev) | CAACAGCGTGCTGGTGCAGTCTACGTTGGTTTC |
| KH61 (W104S_Fwd) | CAGCGCAACATTCCCAGCCTGATTAAAAGCATTC |
| KH62 (W104S_Rev) | GAATGCTTTTAATCAGGCTGGGAATGTTGCGCTG |
| KH63 (A128S_Fwd) | GAAACCAACGTAGACAGCACCAGCACGCTGTTG |
| KH64 (A128S_Rev) | CAACAGCGTGCTGGTGCTGTCTACGTTGGTTTC |
| LP164 (T48C_Fwd) | GTAAGTTTTGATAAAACCTGCTTTCGTTTGCTGCGCG |
| LP165 (T48C_Rev) | CGCGCAGCAAACGAAAGCAGGTTTTATCAAAACTTAC |
| LP168 (K186C_Fwd) | CCATTCCGATAGAACTATGCCGCTCCTATATGGTGTG |
| LP169 (K186C_Rev) | CACACCATATAGGAGCGGCATAGTTCTATCGGAATGG |
| KH35 (T48S_Fwd) | GTTTTGATAAAACCAGCTTTCGGCTGCG |
| KH36 (T48S_Rev) | CGCAGCAAACGAAAGCTGGTTTTATCAAAA |

*Table 3. Continued on next page*

*Table 3. Continued*

| Primer # (name) | Sequence (5′–3′) |
| --- | --- |
| KH39 (K186S_Fwd) | CATTCCGATAGAACTAAGTCGCTCCTATATGGTG |
| KH40 (K186S_Rev) | CACCATATAGGAGCGACTTAGTTCTATCGGAATG |
| KH45 (PhoQ_tetRA_knock-in_Fwd) | GAATAAATTTGCTCGCCATTTTCTGCCGCTGTCGCTGCGGTTAAGACCCACTTTCACA |
| KH46 (PhoQ_tetRA_knock-in_Rev) | CCTCTTTCTGTGTGGGATGCTGTCGGCCAAAAACGACCTCCTAAGCACTTGTCTCCTG |
| KH93 (ST-PhoQ_N-term_Fwd) | ATGAATAAATTTGCTCGCCATTTTC |
| KH94 (ST-PhoQ_N-term_Rev) | TTATTCCTCTTTCTGTGTGGG |
| KH265 (ST-rpoD_Fwd_qRT) | GGGATCAACCAGGTTCAATG |
| KH266 (ST-rpoD_Rev_qRT) | GGACAAACGAGCCTCTTCAG |
| KH269 (ST-pagD_Fwd_qRT) | GTTCAGGCCATTGTTCTGGT |
| KH270 (ST-pagD_Rev_qRT) | TAATCTGCCTGGCTTGCTTT |
| KH273 (ST-pagO_Fwd_qRT) | CGGGCTTAACTATCGCAATC |
| KH274 (ST-pagO_Rev_qRT) | CAGCAGAAATAAGCGCAGTG |
| KH275 (ST-phoP_Fwd_qRT) | TGCCAGGGAAGCTGATTACT |
| KH276 (ST-phoP_Rev_qRT) | CAGCGGCGTATTAAGGAAAG |
| KH277 (ST-phoN_Fwd_qRT) | CCGGCTTACCGCTATGATAA |
| KH278 (ST-phoN_Rev_qRT) | CGCTTACATCTGCATCCTCA |

(Agilent Technologies, Santa Clara, CA). The resulting mutagenized pBAD24-*phoQ* plasmids were transformed into CS1081 and grown overnight on LB plates containing XP substrate (Sigma 104, Sigma-Aldrich Corp., St. Louis, MO), ampicillin 100 µg•ml$^{-1}$, and 10 mM MgCl$_2$. In the morning, plates were screened for blue colonies indicative of *phoN::TnphoA* alkaline phosphatase activity and PhoQ activation by divalent cation limitation. Approximately, 50,000 colonies were screened. 103 blue colonies were chosen and sequenced, yielding 26 single amino acid substitutions. Mutations identified in the screen were independently engineered in a clean pBAD24-*phoQ* background using the appropriate primers pairs (*Table 3*) and a standard site-directed mutagenesis protocol. Phenotypes were confirmed by alkaline phosphatase activity assays in duplicate on at least three separate occasions.

## Protein expression and purification

The PhoQ PD (strain CS1101) was purified as previously described (*Bader et al., 2005*). PhoQ$^{W104C-A128C}$ PD (strain KH85) was purified from SHuffle T7 Express *E. coli* (NEB, Ipswich, MA). Purification and storage of PhoQ$^{W104C-A128C}$ PD was performed according to the same methods as wild-type PhoQ PD. Disulfide bond formation in PhoQ$^{W104C-A128C}$ PD was confirmed by SDS-PAGE. Briefly, strain CS1101 or KH85 were grown in LB media supplemented with 100 mg•l$^{-1}$ ampicillin for all non-labeling experiments. For $^{15}$N-labeling and NMR experiments, strain CS1101 was grown in MOPS minimal medium supplemented with 100 mg•l$^{-1}$ ampicillin and 1 g•l$^{-1}$ $^{15}$N-ammonium chloride. Expression strains were grown to mid-log phase and IPTG was added to 0.5 mM. Cultures were induced for 4–6 hr, harvested by centrifugation and lysed using a French Pressure Cell. Inclusion bodies were isolated by centrifugation, washed once in 50 mM sodium phosphate pH 8.0 300 mM NaCl, resuspended in 20 mM sodium phosphate pH 8.0 100 mM NaCl 7 M urea, and incubated on ice for 1 hr. Samples were then ultracentrifuged at 50,000 rpm for 30 min. The supernatant was rapidly diluted into ice cold 20 mM sodium phosphate pH 8.0. Samples were filtered and purified using a 5 ml HisTrap HP nickel column (GE Healthcare Bio-Sciences, Pittsburgh, PA) according to standard protocol. Purified protein was then applied to a Superdex-200 gel filtration column (GE Healthcare Bio-Sciences) equilibrated with 20 mM sodium phosphate pH 6.5 150 mM NaCl 0.1 mM EDTA. PhoQ containing fractions were pooled, concentrated to approximately 0.25 mM and stored at −80°C in 10% glycerol. PhoQ$^{W104C-A128C}$ PD was expressed by growing strain KH85 at 37°C to mid log phase in LB medium supplemented with 100 mg•l$^{-1}$ ampicillin. IPTG was added to 0.5 mM and protein expression was maintained overnight at 20°C.

## NMR spectroscopy and analysis

($^1$H, $^{15}$N)-HSQC-NMR spectra of PhoQ PD were collected as a function of pH previously (*Prost et al., 2007*). Briefly, uniformly $^{15}$N- and $^{13}$C-labeled PhoQ PD was prepared to 1.2 mM in 20 mM sodium phosphate buffer pH 6.5, 150 mM NaCl, 20 mM MgCl$_2$, 0.1 mM EDTA, and 10% (vol/vol) D$_2$O. The pH was lowered by approximately 0.5 units at a time by addition of microliter aliquots of 500 mM DCl. HSQC spectra from the pH-titration were collected at pH 6.5, 6.0, 5.5, 4.9, 4.1, and 3.5. Standard triple-resonance experiments were collected a pH 3.5 for assignments. Assignments from pH 3.5 were translated to higher pH conditions by tracking chemical shifts through the titration series. NMR experiments were performed at 25°C on a Bruker DMX 500 MHz spectrometer equipped with a triple-resonance, triple-axis gradient probe. Data were processed and analyzed using the programs NMRPipe/NMRDraw (*Delaglio et al., 1995*) and NMRView (*Johnson and Blevins, 1994*).

To identify regions of the PhoQ PD affected by pH, ($^1$H,$^{15}$N)-HSQC-NMR spectra of the PhoQ PD were compared at pH 6.5 and 3.5. Resonances that experienced a chemical shift perturbation (CSP) greater than 0.08 ppm and/or that broadened beyond detection were considered significantly affected by pH. CSPs were calculated using the formula $((\Delta^1H) + (\Delta^{15}N/5))^{1/2}$. Resonances that did not meet these criteria were considered unaffected. 36 residues within the PhoQ PD could not be unambiguously categorized into these groups because of missing/ambiguous assignments and/or crowding in the spectra.

## Bacterial gene expression from growth in N-minimal media

Wild-type (CS093), *phoQ*$^{W104C-A128C}$ (KH163), and *ΔphoQ* (CS1350) *S. enterica* Typhimurium were grown overnight in N-mm 1 mM MgCl$_2$ pH 7.5. In the morning, the cultures were normalized to OD$_{600}$ 2.0 and diluted 1:50 into fresh N-mm 1 mM MgCl$_2$ pH 7.5 and grown at 37°C, 250 rpm. At approximately OD$_{600}$ 0.2, the cultures were normalized to OD$_{600}$ 0.2•ml$^{-1}$, washed once, and resuspended in 1 ml of either N-mm pH 7.5 1 mM MgCl$_2$, pH 5.5 1 mM MgCl$_2$, pH 7.5 10 μM MgCl$_2$, or pH 7.5 1 mM MgCl$_2$ 5 μg•ml$^{-1}$. The cultures were grown shaking at 37°C, 250 rpm. After 1 hr, the cultures were immediately pelleted at 4°C, the media was aspirated, and placed on ice. RNA was collected using the Trizol Max Bacterial RNA Isolation Kit (Ambion, Thermo Fisher Scientific, Grand Island, NY) and RNeasy mini kit (Qiagen, Netherlands). cDNA was generated using SuperScript III First-Strand (Invitrogen, Thermo Fisher Scientific, Grand Island, NY).

Synthesis Supermix for qRT-PCR (Invitrogen). Quantitative RT-PCR was performed using SYBR GreenER qPCR SuperMix Universal (Invitrogen) and a BioRad CFX96 thermocycler for *S. enterica* Typhimurium *rpoD*, *pagD*, *pagO*, *phoN*, and *phoP* target transcripts using the appropriate qRT primers (*Table 3*). Relative gene expression was determined using the $2^{-\Delta \Delta C_T}$ method (*Livak and Schmittgen, 2001*). *rpoD* was used as the calibrator and gene expression was normalized to *ΔphoQ*.

## Protein crystallization, data collection, and structure determination

The *S. enterica* Typhimurium PhoQ$^{W104C-A128C}$ PD structure (PDB 4UEY) was acquired by crystallizing the purified protein using a Mosquito crystallization robot (TTP Labtech, United Kingdom) and Nextal Classic Suite, Nextal Classic Suite II, Protein complex Suite (Qiagen) and JBScreen Classic HTS II (Jena Bioscience, Germany). The progress of crystallization at 20°C was monitored using a temperature controlled robot (Rock imager system, Formulatrix, Bedford, MA). Crystals appeared after 2 weeks. Optimized crystals of the PhoQ$^{W104C-A128C}$ PD were formed in 0.1 M Bis-Tris pH 6.5 200 mM Magnesium chloride 25% (wt/vol) PEG3350. Crystals of the PhoQ$^{W104C-A128C}$ PD were mounted in nylon loops (Hampton Research, Aliso Viejo, CA) and directly frozen in liquid nitrogen. Diffraction data of the crystals were collected at ALBA synchrotron (BL13 XALOC, Barcelona, Spain). Crystals were kept at 100 K and 200 diffraction images at 1° were recorded on a Pilatus 6M detector (Dectris, Baden, Switzerland). Diffraction data were processed and scaled using the XDS software package (*Kabsch, 2010*). Data were truncated at lower resolution according to the recently defined CC* correlation factor (*Karplus and Diederichs, 2012*). Molecular replacement trials were performed using the program MOLREP and the model of the *S. enterica* Typhimurium PhoQ PD from the PDB databank (PDB 1YAX) (*Cho et al., 2006*; *Vagin and Teplyakov, 2010*). The structure was refined using the PHENIX program package (*Afonine et al., 2012*) after rebuilding the structure in COOT (*Emsley et al., 2010*). Structure details and PDB entries are given in *Table 1*. Model quality was assessed using the Molprobity server (http://molprobity.biochem.duke.edu/).

## Circular dichroism

Prior to CD data collection, purified PhoQ PD and PhoQ$^{W104C-A128C}$ PD were exchanged into 20 mM sodium phosphate buffer pH 5.5 150 mM NaCl 1 mM $MgCl_2$ using a 5 ml HiTrap desalting column (Amersham) and treated with or without an approximate 1000 molar excesses of TCEP hydrochloride (Sigma) pH 5.5 for 4 hr to reduce the disulfide bond formed between W104C and A128C. Following TECP treatment, protein samples were exchanged in to 20 mM sodium phosphate buffer pH 5.5 150 mM NaCl 1 mM $MgCl_2$, with or without 1 mM TCEP and equilibrated overnight at 4˚C. Following buffer exchange and equilibration, protein samples were concentrated and prepared to 17 µM for CD analysis. Disulfide bond reduction was monitored by SDS-PAGE prior to performing CD experiments. All CD data collection was performed on an Aviv model 420 spectrometer fitted with a total fluorescence accessory module and thermoelectric cuvette holder using a 1 mm pathlength quartz cuvette. Wavelength scans were performed for each sample prior to thermal denaturation from 260 to 195 nm at 25˚C, sampling every 1 nm, with a 3 s averaging time per reading. CD-monitored thermal denaturation data was collected at 212 nm, from 25˚C to 95˚C, in 1˚C increments, with a 3 s averaging time per reading, and 30 s temperature equilibration between readings. Raw thermal denaturation data were normalized to give the fraction unfolded protein assuming a two-state denaturation process (Kamal et al., 2002). All CD experiments were reproduced on at least three separate occasions.

## Mouse infections

BALB/c or A/J mice were ordered from Jackson Laboratories and virulence phenotypes for strains of *S. enterica* Typhimurium were determined by competition or single-strain inoculation. Competition experiments were performed similarly to previously described (Freeman et al., 2003). Briefly, cultures of KH111, KH112, KH113, and KH114 were grown overnight in LB media with the appropriate antibiotic and prepared by serial dilution in PBS. The inoculum for IP competition experiments was prepared by equally mixing $2.5 \times 10^5$ cfu of KH111 (strain A) with $2.5 \times 10^5$ cfu of KH112, KH113, or KH114 (strain B) in 2 ml PBS. The inoculum for PO competition experiments was prepared by equally mixing $5 \times 10^7$ cfu of KH111 (strain A) with $5 \times 10^7$ cfu of KH112, KH113, or KH114 (strain B) in 2 ml PBS. 6- to 8-week old female BALB/c mice were administered 0.2 ml of the mixture, for a total inoculation of $1 \times 10^5$ bacteria for IP infections or $5 \times 10^8$ bacteria for PO infections. For PO competition experiments, mice were deprived food for 5 hr prior to administering bacteria by oral gavage. The inoculum was confirmed for each experiment by plating dilutions on LB media supplemented with either 50 µg•ml$^{-1}$ kanamycin or 100 µg•ml$^{-1}$ ampicillin. Mice were euthanized by $CO_2$ asphyxiation at 48-hpi (IP) or 96-hpi (PO) and spleens were harvested and homogenized in PBS. Homogenized spleens were serial diluted and plated on LB media supplemented with either 50 µg•ml$^{-1}$ kanamycin or 100 µg•ml$^{-1}$ ampicillin in order to determine the cfu•ml$^{-1}$ bacterial burden for each strain. The competitive index (CI) for each strain was calculated using the following formula: CI = (strain B cfu•ml$^{-1\ spleen}$/strain A cfu•ml$^{-1\ spleen}$)/(strain B cfu•ml$^{-1\ inoculum}$/strain A cfu•ml$^{-1\ inoculum}$).

For single-strain experiments, cultures of CS093, KH163, and CS1350 were grown overnight in LB media and prepared by serial dilution in PBS. The inoculum was confirmed for each experiment by plating dilutions on LB media. 6- to 8-week old female BALB/c or A/J mice were infected IP with approximately $1 \times 10^3$ cfu in 0.2 ml PBS. Mice were euthanized by $CO_2$ asphyxiation at 48- and 96-hpi and spleens were harvested and homogenized in PBS. Homogenized spleens were serial diluted and plated on LB media in order to determine the cfu•ml$^{-1}$ bacterial burden for each strain. All mouse experiments were performed with IACUC approval.

## Bacterial growth curve

Wild type (CS093), *phoQ$^{W104C-A128C}$* (KH163), and *ΔphoQ* (CS1350) were grown overnight in N-mm pH 7.5 1 mM $MgCl_2$. The following morning, the strains were washed in the appropriate N-mm, normalized, and diluted to 0.05 OD$_{600}$ in either N-mm pH 7.5 or pH 5.5 supplemented with 1 mM $MgCl_2$. The strains were grown in a rolling drum at 37˚C. At the indicated time-points, the bacterial strains were diluted 1:10 in PBS and their OD$_{600}$ was monitored.

## Macrophage growth conditions and infections

Bone marrow was isolated from the femurs of BALB/c mice obtained from Jackson Laboratories and differentiated for 7 days in RPMI 1640 media (Gibco #22400-089, Thermo Fisher Scientific, Grand

Island, NY) supplemented with 10% FBS and L-929 cell supernatant following standard protocols. Following differentiation, bone-marrow derived macrophages were seeded into 24-well plates and incubated overnight. Bone-marrow derived macrophages were infected in triplicate with CS093, KH163, or CS1350 *S. enterica* Typhimurium and bacterial survival determined using a standard gentamicin-protection assay. Briefly, CS093, KH163, and CS1350 were grown overnight in LB media. The following morning, bacterial cultures are washed in PBS and suspended in RPMI 1640 at the appropriate concentration. BALB/c bone-marrow derived macrophages in 24-well plates ($2 \times 10^5$ per well) were washed with PBS and infected in triplicate with CS093, KH163, or CS1350 (M.O.I. of 10) in RPMI 160 supplemented with 10% FBS, synchronized by centrifugation at 1000 rpm for 5 min at RT, and incubated for 30 min. Following incubation, infected macrophage monolayers were washed with PBS, incubated with media supplemented with 100 µg•ml gentamicin$^{-1}$ (Sigma) for 90 min and maintained at 15 µg•ml$^{-1}$ gentamicin for the duration of the experiment. Bacterial intracellular survival was determined by lysing infected macrophage with 1% Triton X-100 in PBS at the indicated time-points and plating serial dilutions on LB media for cfu counting.

## Bacterial gene expression from within infected macrophage

BALB/c bone marrow-derived macrophages were seeded into 6-well plates ($1 \times 10^7$ per well) and infected in triplicate with CS093, KH163, or CS1350 *S. enterica* Typhimurium using a standard gentamicin-protection protocol. 30 min post-infection, extracellular bacteria were harvested, lysed in Max Bacterial Enhancement Reagent (Ambion) and RNA was stabilized with Trizol (Ambion). 4 hr post-infection, media was aspirated, infected macrophages were solubilized in Trizol to stabilize total RNA and triplicates where pooled. Trizol samples were stored at −80°C. RNA was prepared according to the Trizol Reagent protocol, treated with TURBO DNA-free DNase (Ambion), and RNA quality was monitored using a 2200 TapeStation (Agilent Technologies). cDNA was generated using SuperScript III First-Strand Synthesis Supermix for qRT-PCR (Invitrogen). Quantitative RT-PCR was performed using SYBR GreenER qPCR SuperMix Universal (Invitrogen) and a BioRad CFX96 thermocycler for *S. enterica* Typhimurium *rpoD*, *pagD*, *pagO*, *phoN*, and *phoP* target transcripts using the appropriate qRT primers (*Table 3*). Relative gene expression was determined using the $2^{-\Delta \Delta C_T}$ method (*Livak and Schmittgen, 2001*). *rpoD* cDNA generated from extracellular bacteria harvested 30 min post-infection was used as the calibrator.

## Three-dimensional structural analysis

Analysis and modeling of the three-dimensional protein structures was carried out using the PyMOL molecular viewer (*Schrodinger, 2010*).

## Acknowledgements

We would like to thank C Davey, Z Dalebroux, H Kulasekara, A Imhaus, J Fan, and S Matamouros for constructive feedback, A Hajjar for assistance with mouse infections, D Baker and F Parmeggiani for CD spectrometer assistance, and A Rojas at ALBA synchrotron for data collection. This work was funded by grant R01AI030479 from the NIAID to S.I.M.

## Additional information

### Funding

| Funder | Grant reference | Author |
| --- | --- | --- |
| National Institute of Allergy and Infectious Diseases (NIAID) | R01AI030479 | Samuel I Miller |

The funder had no role in study design, data collection and interpretation, or the decision to submit the work for publication.

### Author contributions

KGH, Conception and design, Acquisition of data, Analysis and interpretation of data, Drafting or revising the article, Contributed unpublished essential data or reagents; SPD, Conception and design, Acquisition of data, Analysis and interpretation of data, Drafting or revising the article; ES-V,

KZ, Acquisition of data, Analysis and interpretation of data, Drafting or revising the article; M-PB, LRP, MED, Conception and design, Acquisition of data, Analysis and interpretation of data; KKD, Acquisition of data, Analysis and interpretation of data, Drafting or revising the article, Contributed unpublished essential data or reagents; REK, SIM, Conception and design, Analysis and interpretation of data, Drafting or revising the article

## Ethics

Animal experimentation: This study was performed in strict accordance with the recommendations in the Guide for the Care and Use of Laboratory Animals of the National Institutes of Health. All of the animals were handled according to approved institutional animal care and use committee (IACUC) protocol (2982-02) of the University of Washington.

# Additional files

## Major datasets

The following dataset was generated:

| Author(s) | Year | Dataset title | Dataset ID and/or URL | Database, license, and accessibility information |
|---|---|---|---|---|
| Sancho-Vaello E, Hicks KG, Miller SI, Zeth K | 2015 | Structure of the periplasmic domain PhoQ double mutant (W104C-A128C) | http://www.rcsb.org/pdb/search/structidSearch.do?structureId=4UEY | Publicly available at RCSB Protein Data Bank (Accession no: 4UEY). |

The following previously published datasets were used:

| Author(s) | Year | Dataset title | Dataset ID and/or URL | Database, license, and accessibility information |
|---|---|---|---|---|
| Cho US, Bader MW, Amaya MF, Daley ME, Klevit RE, Miller SI, Xu W | 2006 | Cystal structure Analysis of *S. typhimurium* PhoQ sensor domain with Calcium | http://www.rcsb.org/pdb/explore/explore.do?structureId=1YAX | Publicly available at RCSB Protein Data Bank (Accession no: 1YAX). |
| Cheung J, Bingman CA, Reyngold M, Hendrickson WA, Waldburger CD | 2008 | Crystal Structure of the *E. coli* PhoQ Sensor Domain | http://www.rcsb.org/pdb/explore/explore.do?structureId=3BQ8 | Publicly available at RCSB Protein Data Bank (Accession no: 3BQ8). |

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
