## [Decision Letter]

Thank you for sending your work entitled “Acidic pH and divalent cation sensing by PhoQ are dispensable for systemic salmonellae virulence” for consideration at *eLife*. Your article has been favorably evaluated by Richard Losick (Senior editor) and three reviewers, one of whom is a member of our Board of Reviewing Editors.

The Reviewing editor and the other reviewers discussed their comments before we reached this decision, and the Reviewing editor has assembled the following comments to help you prepare a revised submission.

In this study, Hicks et al. first performed genetic mutagenesis analyses and NMR structural analyses of PhoQ PD to reveal the mechanism of PhoQ sensing of acid pH. They identified α4 and α5 and their interactions with the α/β core as the pH-responsive structural element; residues with these structural regions experience ionization and dynamic conformational changes upon transition to acid pH condition. Bases on the proposed structural mechanism, the authors then designed a PhoQ^W104C-A128C^ mutant that has an internal disulfide bond expected to restrict the conformal changes of α4/α5 required for sensing acidic pH. Significantly, the mutant does not respond to acidic pH and divalent limitation-induced PhoQ activation by still can mediate cationic antimicrobial peptide (CAMP) activation of PhoQ. This is particularly important as it is for the first time to separate the function of PhoQ sensing of CAMP from sensing the acidic pH and divalent cation limitation. When expressed in *Salmonella*, the authors provided strong data demonstrating that the PhoQ^W104C-A128C^ mutant behaved similarly as the WT protein in bacterial replication both in cultured macrophages as well as in infected animals. These data potentially resolve a longstanding puzzle about the in vivo function of the PhoP-PhoQ two-component system, i.e. regulating bacterial virulence mainly by responding to CAMP rather than acidification of the *Salmonella*-containing phagosome and divalent cation limitation. PhoQ is one of the major regulators of pathogenesis in *Salmonella*, and is also one of the most well studied histidine kinases. The results presented in this paper are interesting and important, and will be of broad interest. The authors should consider the following points to revise the manuscript.

Major comments: 1) A major part of the paper depends on the claim that PhoQ^W104C-A128C^ is insensitive to acid pH and divalent cations and sensitive to CAMPs. However, the only evidence is based on a single *phoN* reporter (Figure 4). Given the importance of this property of the PhoQ mutant for the rest of the paper, a little more evidence should be presented to support the claim. Similar data showing that pH and divalent cations do not affect PhoQ-dependent transcription (but CAMPs do) for *phoP* and some other well-studied gene should be shown.

2) CAMP directly competes with divalent cation for binding sites in the PhoQ PD acidic patch, suggesting a model that the CAMP activates PhoQ by disrupting salt-bridges with the inner membrane. If the CAMP functions through a similar mechanism as the divalent cation in inducing the conformational changes of PhoQ (Figure 7), why the PhoQ^W104C-A128C^ mutant that lost the response to divalent cation limitation can still respond to the CAMP. The authors should clarify this in their presentation.

3) The PhoQ PD can directly interact with the CAMP via the acidic patch (5). Is it possible to measure the binding constant and kinetics? If so, the PhoQ^W104C-A128C^ mutant should exhibit a different behavior, for example on an ITC binding assay, due to its restricted motion in α4/α5 structure. Related to the potential mechanism of PhoQ PD sensing of the CAMP, the authors should mention whether there is a functional PhoQ mutant deficient in responding to the CAMP that they are aware of (what about the 14 mutants shown in Figure 1?).

4) In the Discussion, it is stated: “Further experiments will need to be performed to examine the contribution of acidic pH and divalent cation sensing to PhoQ-mediated bacterial survival during transition from the intestinal tract to systemic environments and determine if ‘bacterial innate immunity’ or the recognition of multiple mammalian signals is redundant.” The authors should test their PhoQ mutant in competition with wild-type in oral infection of mice. Either outcome (WT outcompetes the mutant or they have the same CI) would be interesting, and this experiment is important for conveying the full significance of the authors’ work and should be straightforward to do.

---

## [Author Response]

*1) A major part of the paper depends on the claim that PhoQ*^*W104C-A128C*^
*is insensitive to acid pH and divalent cations and sensitive to CAMPs. However, the only evidence is based on a single* phoN *reporter (*Figure 4*). Given the importance of this property of the PhoQ mutant for the rest of the paper, a little more evidence should be presented to support the claim. Similar data showing that pH and divalent cations do not affect PhoQ-dependent transcription (but CAMPs do) for* phoP *and some other well studied gene should be shown*.

To provide further evidence that PhoQ^W104C-A128^ is inhibited for activation by acidic pH and divalent cation limitation, but remains responsive to CAMP, qRT-PCR was performed on the PhoQ-dependent transcripts *pagD*, *pagO*, *phoN*, and *phoP* from bacteria grown in various activating N-minimal media conditions. Similar to the *phoN::TnphoA* reporter, the PhoQ-dependent transcripts analyzed displayed decreased transcription in the *phoQ*^*W104C-A128C*^ background when exposed to acidic pH and divalent cation limitation, but CAMP-mediated transcription was similar to wild type. Although *pagD*, *pagO*, and *phoN* are highly upregulated in a PhoQ-dependent manner, in our hands, *phoP* transcription displayed only slight to moderate induction. We have included these qRT-PCR results in the subsection “A disulfide bond between α-helices 2 & 4 within the PhoQ PD inhibits activation by acidic pH and divalent cation limitation, but does not restrict activation by CAMP” and as Figure 3—figure supplement 2.

*2) CAMP directly competes with divalent cation for binding sites in the PhoQ PD acidic patch, suggesting a model that the CAMP activates PhoQ by disrupting salt-bridges with the inner membrane. If the CAMP functions through a similar mechanism as the divalent cation in inducing the conformational changes of PhoQ (Figure 7), why the PhoQ*^*W104C-A128C*^
*mutant that lost the response to divalent cation limitation can still respond to the CAMP. The authors should clarify this in their presentation*.

The observations that PhoQ^W104C-A128C^ responds to CAMP and not divalent cation limitation suggests that disruption of salt-bridges between the PD and the inner membrane may not be the entire mechanism by which CAMP activates PhoQ. Furthermore, our findings are not mutually exclusive from those of [5]. Perhaps CAMP activates PhoQ in a series of steps beginning with the displacement of divalent cations at the acidic patch, followed by interactions with the adjacent periplasmic domain or transmembrane regions or disruption of local phospholipid packing. Further investigation will be required to determine the mechanism by which CAMP activates PhoQ. We have clarified this in the Discussion by suggesting possible mechanisms by which CAMP activates PhoQ.

*3) The PhoQ PD can directly interact with the CAMP via the acidic patch (*[5]*). Is it possible to measure the binding constant and kinetics? If so, the PhoQ*^*W104C-A128C*^
*mutant should exhibit a different behavior, for example on an ITC binding assay, due to its restricted motion in α4/α5 structure. Related to the potential mechanism of PhoQ PD sensing of the CAMP, the authors should mention whether there is a functional PhoQ mutant deficient in responding to the CAMP that they are aware of (what about the 14 mutants shown in*
Figure 1*?)*.

We are unclear why the reviewers wanted us to compare the CAMP binding affinities of wild type and PhoQ^W104C-A128C^. We have previously measured cationic peptide binding to the purified periplasmic domain and are not sure why we would reevaluate this given our observation that CAMP activates wild type and *phoQ*^*W104C-A128C*^ similarly, suggesting that the binding interactions are roughly equivalent. Additionally, the known CAMP and divalent cation binding sites in the acidic patch were not significantly altered in the PhoQ^W104C-A128C^ PD structure. Measuring CAMP binding affinity of wild type and PhoQ^W104C-A128C^ should ideally be performed in the context of a membrane, as soluble domains may not necessarily reflect physiological binding interactions. Such experiments seem outside the scope of this study, would be difficult to develop and perform, and would delay timely publication and dissemination of our results to interested investigators.

We have not identified a mutation that exclusively abolishes PhoQ activation by CAMP without perturbing the response to acidic pH and/or divalent cation limitation. [5] defined a mutation in the acidic patch (T156K E184K) that reduces PhoQ activation by CAMP; however, recent reevaluation has revealed that this mutant also displays perturbed activation by divalent cation limitation, which was expected since it forms a metal binding site (unpublished observations). Extensive point-mutations in the PhoQ PD acidic patch can lead to reduced activation by CAMP, but activation by acidic pH and divalent cation limitation are also perturbed (unpublished observations). Further significant investigation is required to elucidate the mechanism by which CAMP activates PhoQ.

*4) In the Discussion, it is stated: “Further experiments will need to be performed to examine the contribution of acidic pH and divalent cation sensing to PhoQ-mediated bacterial survival during transition from the intestinal tract to systemic environments and determine if ‘bacterial innate immunity’ or the recognition of multiple mammalian signals is redundant.” The authors should test their PhoQ mutant in competition with wild-type in oral infection of mice. Either outcome (WT outcompetes the mutant or they have the same CI) would be interesting, and this experiment is important for conveying the full significance of the authors’ work and should be straightforward to do*.

A competition between wild type and *phoQ*^*W104C-A128C*^ via oral infection was a great suggestion and provides broader significance to our findings. As requested, peroral competition infections in BALB/c mice were performed similar to our intraperitoneal competition infections to determine whether acidic pH and/or divalent cation sensing by PhoQ is important in host sites prior to systemic spread. Although more variable due to the nature of the experiment, we observed competitive indices between *wt*/*wt* and *wt*/*phoQ*^*W104C-A128*^ in the spleen similar to the IP competitions. *ΔphoQ* had a lower mean competitive index, as expected. These results suggest that acidic pH and divalent cation sensing by PhoQ are not required for *S. enterica* Typhimurium virulence via an oral infection route. Data concerning our PO competitions is presented in the Results section (subsection headed “*Salmonella* strains with the *phoQ*^*W104C-A128C*^ allele are competent for survival during systemic virulence in mice and within cultured macrophage”) and in Figure 5—figure supplement 1.